# Potential of Tamarind Shell Extract against Oxidative Stress In Vivo and In Vitro

**DOI:** 10.3390/molecules28041885

**Published:** 2023-02-16

**Authors:** Weixi Li, Rongping Huang, Shaocong Han, Xiyou Li, Haibiao Gong, Qiongyi Zhang, Changyu Yan, Yifang Li, Rongrong He

**Affiliations:** 1College of Traditional Chinese Medicine, Yunnan University of Chinese Medicine, Kunming 650500, China; 2Guangdong Engineering Research Center of Chinese Medicine & Disease Susceptibility, Jinan University, Guangzhou 510632, China

**Keywords:** tamarin shell extract, flavonoids, oxidative stress, free radicals scavenging, AAPH, *tert*-butyl hydroperoxide, zebrafish

## Abstract

Tamarind shell is rich in flavonoids and exhibits good biological activities. In this study, we aimed to analyze the chemical composition of tamarind shell extract (TSE), and to investigate antioxidant capacity of TSE in vitro and in vivo. The tamarind shells were extracted with 95% ethanol refluxing extraction, and chemical constituents were determined by ultra-performance chromatography–electrospray tandem mass spectrometry (UPLC-MS/MS). The free radical scavenging activity of TSE in vitro was evaluated using the oxygen radical absorbance capacity (ORAC) method. The antioxidative effects of TSE were further assessed in 2,2-azobis (2-amidinopropane) dihydrochloride (AAPH)-stimulated ADTC5 cells and *tert*-butyl hydroperoxide (*t*-BHP)-exposed zebrafish. A total of eight flavonoids were detected in TSE, including (+)-catechin, taxifolin, myricetin, eriodictyol, luteolin, morin, apigenin, and naringenin, with the contents of 5.287, 8.419, 4.042, 6.583, 3.421, 4.651, 0.2027, and 0.6234 mg/g, respectively. The ORAC assay revealed TSE and these flavonoids had strong free radical scavenging activity in vitro. In addition, TSE significantly decreased the ROS and MDA levels but restored the SOD activity in AAPH-treated ATDC5 cells and *t*-BHP-exposed zebrafish. The flavonoids also showed excellent antioxidative activities against oxidative damage in ATDC5 cells and zebrafish. Overall, the study suggests the free radical scavenging capacity and antioxidant potential of TSE and its primary flavonoids in vitro and in vivo and will provide a theoretical basis for the development and utilization of tamarind shell.

## 1. Introduction

Balance of redox is well maintained under physiological conditions, which is essential for homeostasis of body. However, oxidative stress or oxidative damage, an imbalance state of redox, can occur under some pathological conditions or under stimulation by exogenous harmful substances [1]. Oxidative stress is characterized by the excessive production of reactive oxygen species (ROS)/reactive nitrogen species (RNS) in the body, including superoxide anion, hydrogen peroxide, hydroxyl radicals, nitric oxide and nitrogen dioxide [2]. Although ROS/RNS is essential for cell survival, its abnormal accumulation might directly damage intracellular biomacromolecules such as nucleic acid and proteins or initiates various signaling pathways, ultimately causing cellular disorder and even cell death [3]. Numerous studies have shown that ROS could induce cell apoptosis by DNA damage and cell ferroptosis by trigger lipid peroxidation [4]. Up to now, diverse studies have demonstrated that oxidative stress causes aging and stress-provoked diseases [5,6]. Therefore, eliminating excessive ROS is of great significance for homeostasis recovery. The major antioxidant defense system of body includes enzymes such as superoxide dismutase (SOD), catalase (CAT) and glutathione peroxidase (GPx), and nonenzymatic antioxidants, such as glutathione (GSH), which can protect against oxidative stress [7]. In addition, exogenous antioxidant supplementation is an important strategy to eliminate free radicals and prevent or treat related diseases [8]. 

*Tamarindus indica Linn (Leguminosae)*, also known as sour beans, sour plum, “Ma hang Huang (Dai language), “Mu han” (Dai language), tamarind, etc. (Subfamily Caesalpiniodeae, family Fabaceae), is a rich source of antioxidant molecules [9]. *Tamarindus indica Linn* is a subtropical tree plant, originating in Africa, that is mainly distributed in Taiwan, Guangdong, Guangxi, Yunnan in China [10]. Tamarind is not only a fruit but also poses significant medicinal and industrial value [11]. The tamarind fruit is palatable due to its sweet and sour taste. Tamarind fruit is abundant in nutrients, such as fructose, glucose, cellulose [12] and amino acids, which can promote saliva secretion, stagnate digestion and whet the appetite [13,14]. Tamarind fruits can be consumed directly or processed into tamarind fruit juice beverages, jelly, pastries and other foods [15]. The tamarind seed polysaccharide (TSP), a main component of tamarind seed extract, has been widely applied in food and pharmaceutical industries [16]. Moreover, the seed coat extract of tamarind also contains a plethora of phytochemicals, mainly polyphenols, flavonoids and polysaccharides [17,18,19], which can be used to treat various human diseases [20]. Nakchat’s [21] research revealed that the seed coat extract of tamarin could eliminate ROS, increase the level of GSH and the expression of antioxidant enzymes SOD and CAT, and reduce oxidative stress-induced damage of human skin fibroblasts. The tamarind leaf extract has good inhibitory activity against *Staphylococcus aureus*, *Pseudomonas aeruginosa* and other bacteria [22]. 

Tamarind shells are the by-product from the consumption and food production of tamarind fruit. In recent years, several studies have found that the tamarind shell extract (TSE) has certain antioxidant [23] and hypoglycemic effects [24,25]. However, a huge number of tamarind shells are usually abandoned as waste with a very low utilization rate. Furthermore, tamarind shells are thrown into the environment, which will lead to water and soil pollution. Hence, the study of chemical composition and biological activity of tamarind shells will be helpful for promoting its utilization for higher additional value [26]. Previous studies have reported that TSE is rich in flavonoids and could scavenge free radicals in vitro [23]. However, the antioxidative effects in vivo have not confirmed and the bioactive components in TSE remain unclear. In the present study, we first simultaneously quantified eight flavonoids in TSE by ultra-performance liquid chromatography coupled to electrospray ionization quadrupole orbitrap mass spectrometry (UPLC-MS/MS). The in vitro and in vitro antioxidant activities of TSE and its eight flavonoids were evaluated in AAPH-treated ATDC5 cells and *t*-BHP-treated zebrafish. Hopefully, our work can provide more evidence for application of TSE in the functional food industry. 

## 2. Results

### 2.1. Quantitative Analysis of the Primary Components in TSE

The major components in TSE were identified by UPLC-MS/MS, (+)-catechin, taxifolin, myricetin, eriodictyol, luteolin, morin, apigenin and naringenin were identified by comparing the retention times with corresponding standards and MS data with the reference standards [25]. The contents of 8 flavonoids were listed in Table 1. The concentrations of the 8 flavonoids were calculated precisely according to the respective calibration curves. 

### 2.2. Antioxidative Capacity of TSE and Its Primary Components Evaluated by ORAC

The ORAC assay was used to measure the antioxidative capability of TSE and its primary components. AAPH releases alkyl peroxyl radicals in aqueous solution to attack sodium fluorescein, resulting in its fluorescence attenuation, which could be suppressed in the presence of free radical scavenger [27]. Hence, the fluorescent protection of the component can indicate its antioxidant activity. TSE showed a strong free radical scavenging activity, which inhibited the attenuation of fluorescein signal with a ORAC value of 3527.7 U/g (Figure 1A). The contents of 8 flavonoids in TSE are shown in Figure 1B, and their antioxidative activity also was evaluated by ORAC assay. The 8 flavonoid compounds of TSE inhibited the fluorescence decay of fluorescein sodium, showing a strong antioxidant effect in vitro (Figure 2A,B). The comparison of the antioxidant capacity of eight compounds by the ORAC value was as follows: taxifolin > morin > (+)-catechin > naringin > eriodictyol > luteolin > myricetin > apigenin (Figure 2C). 

### 2.3. Cytoprotective Effects of TSE and the Individual Flavonoid inAAPH-Treated ATDC5 Cells

The cellular toxicity of TSE and its primary components was estimated on the ATDC5 cells by MTT assay. As depicted in Figure 3B, the ATDC5 cells were treated with different concentrations of TSE for 24 h. The negligible cell toxicity was exhibited when the concentration of TSE was lower than 250 μg/mL. All eight flavonoids caused cell death at a dose of 1000 μM, but only leteolin and morin at 100 μM inhibited the cell viability of ATDC5 cells (Figure 3C). Additionally, the viability of AAPH-stimulated ATDC5 cells was significantly and effectively reduced when the dose of AAPH exceeds 156.3 μM (*p* < 0.001) (Figure 3A). However, after being treated with TSE (10–80 μg/mL), the viability of AAPH-stimulated ATDC5 cells was dose-dependently improved (Figure 3D) (*p* < 0.05). Likewise, except luteolin, with the treatment of the other flavonoids in TSE, the survival rate of AAPH-treated ATDC5 cells was remarkably increased at concentration of 100~200 μM (Figure 3E) (*p* < 0.001~0.05). Furthermore, luteolin showed a prominent cytoprotective effect at concentration of 6.25~50 μM (Figure 3E) (*p* < 0.01~0.05).

### 2.4. Effects of TSE and Flavonoids on ROS Level in AAPH-Treated ATDC5 Cells 

The intracellular ROS generation was assessed via detecting the fluorescent intensity of DCF in AAPH treated ATDC5 cells. The ATDC5 cells were treated with different concentrations of AAPH to evaluate AAPH-induced oxidative stress in ATDC5 cells. Compared with the control, 40 mM AAPH significantly increased the intracellular ROS level (*p* < 0.001) one hour later, without any obvious damages to the cells (Figure 4A). Therefore, 40 mM AAPH was used to induce oxidative stress in the ATDC5 cells. As shown in Figure 4B, TSE decreased the ROS production in the AAPH-treated ATDC5 cells in a concentration dependent manner. The eight flavonoids of TSE effectively suppressed the AAPH-induced ROS generation as well (Figure 5). Overall, TSE and its primary components attenuated the intracellular ROS level. Interestingly, inhibitory effect of luteolin at 100 μM for 24 h on ATDC5 cell viability was observed (Figure 3C). In DCF assay, ATDC5 cells were pretreated with luteolin for 2 h and then with AAPH for addition 1 h. Hence, luteolin has intracellular antioxidant activity to inhibit the increase in ROS induced by AAPH in the short term, but stimulation for a longer time can lead to cytotoxicity through other underlying mechanisms. 

### 2.5. Effects of TSE and the 8 Flavonoids on Intracellular MDA Content and SOD Activity in AAPH-Treated ATDC5 Cells

According to Figure 6A,B, the SOD activity significantly decreased (*p* < 0.05) while MDA content increased (*p* < 0.001) in the AAPH group compared with the control group. However, TSE (50, 100, 200 μg/mL) groups showed a significant enhancement in SOD activity compared to the AAPH group (*p* < 0.05). The MDA content was lowered remarkably by TSE administration (50, 100, 200 μg/mL) (*p* < 0.01). In addition, all the flavonoids, except luteolin with a slight effect, could also significantly improve the SOD activity in stressed cells (*p* < 0.01) (Figure 6C). Among eight flavonoid compounds from TSE, eriodictyol, morin, myricetin, and taxifolin obviously reduced the MDA content (*p* < 0.01) (Figure 6D). These results indicated that TSE enhanced the activity of antioxidant enzyme SOD and suppressed the production of lipid peroxidation product MDA to protect the ATDC5 cell against AAPH-induced oxidative stress.

### 2.6. Effects of TSE and Its Components on t-BHP-Induced Oxidative Stress in Zebrafish

The antioxidative effects of TSE in vivo were estimated in *t*-BHP-stressed zebrafish. Similarly, DCFH-DA probe was used to label ROS in zebrafish. The fluorescence change of DCF was observed in a fluorescence microscope. As depicted in Figure 7A,B. The fluorescence intensity in zebrafish treated with 2 mM *t*-BHP was greatly stronger than the control (*p* < 0.01), suggesting that a large amount of ROS was produced. In addition, we also monitored ROS levels in zebrafish after DCFH-DA staining using a fluorescent microplate reader. In TSE (5, 10, 20 μg/mL)-treated group, the fluorescence intensity was significantly decreased, indicating the ROS production was inhibited (*p* < 0.05) (Figure 7C,D). This result confirmed that TSE possessed robust antioxidant capacity in vivo. Furthermore, the effects of the 8 primary flavonoids of TSE on *t*-BHP-induced oxidative stress in zebrafish were detected. Compared with the *t*-BHP group, the ROS generation in zebrafish declined after treatment with these compounds (Figure 8A–D). Notably, eriodictyol, morin and myricetin exhibited a much stronger antioxidative effect in zebrafish compared to other compounds.

Additionally, total SOD activity and MDA content were detected in *t*-BHP stimulated zebrafish treated with TSE and its primary components. The MDA level elevated from 0.77 μmol/g protein to 1.21 μmol/g protein (*p* < 0.001) and SOD activity markedly decreased (*p* < 0.01) in zebrafish loaded with *t*-BHP (Figure 7E,F). By contrast, TSE (5–20 μg/mL) lowered the MDA content (*p* < 0.05) and reversed the SOD activity (*p* < 0.01) (Figure 7E,F). Moreover, administration of all eight flavonoids (10 μM) obviously lowered MDA content (*p* < 0.001) (Figure 8E). Among them, eriodictyol, morin and myricetin, exhibited better inhibitory activities. At the same time, SOD activities in stressed zebrafish were improved to varying degrees when treated with the 8 flavonoids (Figure 8F). Eriodictyol, morin, taxifolin and apigenin had better ameliorative effects on SOD activities than other compounds. Results above suggested that TSE and its flavonoid compounds mitigated the oxidative stress in *t*-BHP-exposed zebrafish.

## 3. Discussion

Though a previous study has reported that the tamarind shell contains flavonoids and has the ability to scavenge free radicals in vitro [23], these flavonoids have not been identified and quantified and whether they account for antioxidant activity of TSE remains unproven. Furthermore, there is no report about antioxidation of tamarind shell in vivo. In this present work, a total of 8 flavonoids were detected in the TSE by LC-MS/MS, including (+)-catechin (5.287 mg/g), taxifolin (8.419 mg/g), myricetin (4.024 mg/g), eriodictyol (6.583 mg/g), luteolin (3.421 mg/g), morin (4.651 mg/g), apigenin (0.203 mg/g), naringenin (0.623 mg/g) (Table 1). Our study confirmed the antioxidant potential of tamarind shells in vitro and in vivo due to these active compounds, which is expected to enhance the utilization of tamarind shells as antioxidant supplements for higher additional value, and to reduce the cost of handling tamarind shells as waste which may cause environmental pollution.

It has been widely accepted that oxidative stress caused by the imbalance of generation and detoxification of free radicals can lead to aging [28], and diverse diseases such as diabetic retinopathy [29], Alzheimer’s disease and Parkinson’s disease [30]. Therefore, finding more effective antioxidants against oxidative stress damage is essential [31]. Zebrafish have been an important preclinical animal model for genetics, drug discovery and developmental biology, due to physiological and genetic similarity to mammals [32]. In this work, *t*-BHP, a standard pro-oxidant molecule, was used as a stimulus to induce zebrafish oxidative stress. Zebrafish exposed to *t*-BHP exhibited a significant increase in the level of MDA, an end product of lipid peroxidation. SOD is a major antioxidant defense enzyme within cells, which catalyzes the dismutation of superoxide anions to oxygen and hydrogen peroxide [33]. Total SOD activity was significantly decreased in *t*-BHP treated zebrafish. The results suggested that *t*-BHP caused oxidative stress in zebrafish. However, TSE was found to attenuate the oxidative stress within zebrafish loaded with *t*-BHP, as reflected by the recovered SOD activity and reduced MDA content. Using DCFH-DA probe, the ROS generation in zebrafish was determined according to the changes of fluorescence intensity of DCF. Excess ROS was produced in *t*-BHP stimulated zebrafish, but TSE remarkably suppressed the over production of ROS (Figure 7). Similar results were observed in AAPH-induced oxidative stress in ADTC5 cells (Figure 4 and Figure 6). All of the above suggested that TSE possessed strong antioxidative effects in vivo and in vitro. The antioxidant capacity was further confirmed by ORAC assay. TSE had direct free radical clearance activity with an obvious dose-dependent effect (Figure 1).

To further investigate the bioactive components of TSE, the antioxidative effects of the primary flavonoids in TSE were evaluated in vivo and in vitro. The antioxidant capacity of these compounds varied, despite the similarity of chemical structures. Based on the ORAC assays, taxifolin and morin had the best free radical scavenging ability, (+)-catechin, naringenin and eriodictyol as followed. Flavonoids are natural compounds with a 2-phenylchromone structure. The free radical scavenging activity of flavonoids is related to the number and position of the hydroxyl groups in their structures [34]. 

All eight flavonoids had antioxidative effects against AAPH-induced oxidative stress in the ATDC5 cells. However, the flavonoids did not work in the same way. Eight compounds weakened the inhibition of AAPH on ATDC5 cell viability, eliminated the excessive ROS and enhanced the SOD activity in AAPH-treated cells, but only eriodictyol, morin, taxifolin and myricetin had the ability to lower the MDA level (Figure 2, Figure 3, Figure 5 and Figure 6). In addition, the antioxidant activities of eight compounds were studied against the *t*-BHP provoked oxidative stress in zebrafish. All of flavonoids exhibited negligible toxicity in zebrafish when the concentration at 10 μM (Data not shown). The results were largely consistent with the cell experiments. The elevated ROS level in the *t*-BHP-stimulated zebrafish were significantly reduced by the flavonoids. All compounds alleviated the lipid peroxidation, as reflected by the noticeable reduction of MDA. Eriodictyol, morin, taxifolin and apigenin also rehabilitated the SOD activity altered by *t*-BHP (Figure 8).

It has been found that taxifolin have effect of treatment on primary malignant brain tumor by inhibited mTOR/PI3K, promoted autophagy and suppressed lipid synthesis in GBM [35]. In addition, taxifolin also have a cardioprotective effect on diabetes cardiomyopathy by inhibiting oxidative stress and cardiomyocyte apoptosis [36]. Eriodictyol has strong potential of anti-cancer [37,38], antioxidant [39,40], anti-inflammatory [41,42] and so on. In our study, taxifolin and eriodictyol are the most abundant ingredients in TSE and exhibited much stronger antioxidant activities among the eight compounds in vitro and in vivo. Eriodicytol displayed the best antioxidative effect both in *t*-BHP exposure zebrafish and AAPH treated ATDC5 cells. Nevertheless, the ORAC value of eriodictyol was lower than taxifolin. As mentioned above, both taxifolin and eridictyol are flavanols, and their abilities to scavenge the free radicals in vitro is proportional to the number of phenolic hydroxyl groups [34]. The number of phenolic hydroxyl groups in taxifolin is more than eriodictyol, so the ability to scavenge free radicals in vitro is stronger than eriodictyol. However, the additional phenolic hydroxyl group might influence the lipophilicity of taxifolin and the antioxidative effect *in vivo* [43].

## 4. Materials and Methods

### 4.1. Materials

#### 4.1.1. Plant Materials

*Tamarindus indica Linn* was obtained from Jinghong, Xishuangbanna of Yunnan province in China (21°59′35.0″ N; 100°51′29.5″ E), and authenticated by Professor Ya-qiong Li, Yunnan University of Chinese Medicine. Tamarind, about 10–15 m, blooms in May and August. The fruit period is from December to May the next year, and the pods are cylindrical and brown.

#### 4.1.2. Chemicals

(+)-catechin, eriodictyol, luteolin, morin, apigenin, naringenin, taxifolin, and myricetin of ≥98% purity was purchased from Chengdu Desite Bio-Technology (Chengdu, China) and Baishun Biotechnology (Shanghai, China). 2,2′-Azobis(2-methylpropionamidine) dihydrochloride (AAPH, A101386) and N-Acetyl-L-cysteine (NAC) were obtained from Aladdin Biochemical Technology (Shanghai, China). Sodium fluorescein and 6-hydroxy- 2,5,7,8-tetramethylchroman-2-carboxylic acid (Trolox) were bought from Wako Pure Chemical (Osaka, Japan). MTT, Lipid Peroxidation (MDA) Assay Kit (S0131M), Total Superoxide Dismutase (SOD) Assay Kit with WST-8 (S0101M), Reactive Oxygen Species (ROS) Assay Kit, and BCA Protein Assay Kit were purchased from Beyotime Biotechnology (Shanghai, China).

#### 4.1.3. Zebrafish and Husbandry

Zebrafish, a wild-type AB strain were procured from the China zebrafish resource center. All the zebrafish used in current study are housed in an AAALAC-accredited facility. Adult zebrafish were allowed to breed in the zebrafish breeding system (Beijing Aisheng, Beijing, China), with a cycle of 14 h light and 10 h darkness to maintain the light rhythms. The embryos were collected and bred with the zebrafish embryo culture medium in an incubator at 28 °C. The medium contained 59.88 mM NaCl, 2.281 mM NaHCO_3_, 0.6707 mM KCl and 1.059 mM CaCl_2_. All experiments were carried out according to standardized protocols and approved by the Animal Care and Use Committee of Yunnan University of Chinese Medicine.

### 4.2. Methods

#### 4.2.1. Preparing the Tamarind Shell Extract (TSE)

The tamarind shells were separated from the fresh tamarind fruits. The shells were dried for 12 h by Air blast dryer at 50 °C, then pulverized and passed through a 100-mesh sieve. Accurately 500 g of tamarind shell fine powder was mixed with 1500 mL of 95% ethanol, followed by heating reflux twice for 3 h. Later, the separation was precipitated, and the solvent was removed from the extract under reduced pressure by a rotary vacuum evaporator. The extract was freeze-dried and stored at 4 °C until further use.

#### 4.2.2. Quantitative Analysis of Flavonoids in TSE

A total of eight flavonoids ((+)-catechin, taxifolin, myricetin, eriodictyol, luteolin, morin, apigenin and naringenin) in TSE were quantified according to the method of our previous study [25] using the UPLC-MS/MS (Thermo Fisher Scientific, Waltham, MA, USA). Shortly, TSE was solved in water and then filtered through a 0.45 μm nylon 6–6 filter and analyzed. The analysis was carried out by UPLC (Ultimate 3000 DGLC, Thermos Fisher Scientific, USA) equipped with a quadrupole orbitrap mass spectrometry (Q-Exactive) with an ESI source. HSS T3 C18 (100 × 2.1 mm, Waters Acquity) was employed for chromatographic separation. The mobile phase consisted of acetonitrile (Solvent A) and 0.1% aqueous formic acid (Solvent B). The analysis was performed under a gradient program of 0–1 min—5% B, 1–1.5 min—10% B, 1.5–4.5 min—13% B, 4.5–15 min—17% B, 15–20 min—27% B, 20–20.5 min—95% B, 20.5–23 min—95% B, 23–23.2 min—5% B, 23.2–28 min—5% B. The flow rate was 0.3 mL/min, the injection volume was 2 μL and the column temperature was maintained at 40 °C. Mass spectra were obtained in both positive and negative ionization modes. The capillary voltage was 3.0 kV with a source temperature of 320 °C. The collision energy was 20, 30, 40 eV. Quantification of the analytes were calculated by standard calibration curves. All the data were obtained as average values in triplicate experiments.

#### 4.2.3. ORAC Assay

The antioxidative capability of TSE and its primary flavonoids was determined using the ORAC assay according to the previously described method [44,45]. Both Trolox and AAPH were dissolved in potassium phosphate buffer (PBS) immediately before the ORAC assay. The reaction mixture was prepared with 20 μL of 75 mmol L^−1^ PBS, 20 μL TSE or eight flavonoids solution and 20 μL fluorescein sodium in the presence of 140 μL AAPH. The ORAC procedure was carried out on a Synergy H1 fluorescence microplate reader (BioTek, Winooski, VT, USA) with an excitation/emission wavelength of 485/527 nm. Briefly, fluorescein sodium was served as the substrate. AAPH was used to generated peroxyl radicals. The results were calculated based on the area of under the fluorescence decay curve with the Trolox as a standard. All results were expressed as ORAC value which is defined as a protection area under curve of one Trolox unit.

#### 4.2.4. Cell Culture 

The ATDC5 Mouse chondrogenic cells were purchased from Otwo Biotech (Shenzhen, China) and cultured in DMEM/F12-Dulbecco’s Modified Eagle Medium supplemented with 10% fetal bovine serum (FBS, Gibco, Shanghai, China) and 1% penicillin-streptomycin in an incubator at 37 °C with 5% CO_2_. 

#### 4.2.5. MTT Assay

ATDC5 cells were seeded into a 96-well plate at a density of 5 × 10^3^ cells/well over night. Then, the cells were treated with indicated concentration of TSE for 24 h. Next, AAPH (400 μM) was added and incubated for 24 h. The cell viability was detected by MTT assay. After treatment, 10 μL of the MTT solution (5 mg/mL) was added to each well for further incubation for 3 h at 37 °C. The formazan crystals were dissolved with 150 μL DMSO. The absorbance was determined at 570 nm. Cell viability was expressed as the percentage of MTT reduction, the absorbance of control cells was assumed as 100%. In addition, the cytoprotective effects of the eight flavonoids on AAPH-treated ATDC5 cells were also evaluated by MTT assay. 

#### 4.2.6. Measurement of Intracellular Antioxidant Capacity 

The cellular antioxidant activity (CAA) assay was utilized to measure the antioxidant capacity in cell culture of TSE and its main components. The CAA assay was conducted according to Wolfe and Liu with slight modification [46]. In short, ATDC5 cells at a density of 5 × 10^5^ cells/well were seeded and cultured in a 6-well plate at 37 °C with 5% CO_2_ for 24 h. Cells were treated with NAC (2 mM), TSE (50, 100, 200 μg/mL) and flavonoid components (50, 100, 200 μM) and incubated for 2 h, and then added with AAPH (40 mM). After 1 h of treatment, cells were collected and incubated with 10 μM DCFH-DA at 37 °C for 30 min. Then, the cells were washed twice with PBS and immediately analyzed by a Beckman Coulter Epics XL flow cytometer equipped with Expo32 ADC with an excitation/emission filter pair of 485/538 nm. Antioxidant ability was expressed as CAA units by calculating the difference in area under the curve between tested samples and control wells. 

#### 4.2.7. Measurement of SOD Activity and MDA Content in ATDC5 Cells and Zebrafish

Treated cells were washed twice in cold PBS and total proteins were extracted with 100 μL lysis solution for 15 min. The zebrafish were homogenized in 100 μL PBS (pH = 7.4). Lysates or homogenates were centrifuged at 12,000 rpm for 10 min. The protein content in supernatants were quantified with BCA Protein Assay [47]. The SOD activity and MDA content were assessed using commercial kits (Beyotime Biotechnology, Shanghai, China) according to the manufacturer’s instructions. In the detection method of SOD activity, the superoxide anion, produced from the reaction of xanthine with xanthine oxidase (XOD) system, could convert WST-8 to water-soluble formazan dye with a maximum absorption at 450 nm. SOD can inhibit this reaction by consuming superoxide anion. Hence, the enzyme activity of SOD can be calculated by colorimetric analysis. One activity unit (U) indicates the 50% of the inhibition of the reaction. Finally, the activities of SOD of lysate samples were standardized with protein contents. MDA can react with thiobarbituric acid (TBA) to form a red MDA-TBA adduct which has maximum absorption at 535 nm. Therefore, the MDA content were detected by analyzing the level of MDA-TBA adduct. Absorbance was obtained by an automatic microplate reader (Biotek Synergy 4) [48,49].

#### 4.2.8. Waterborne Exposure of Zebrafish

The protective effects of TSE and its major flavonoids against *t*-BHP induced oxidative stress in zebrafish was evaluated using the method described by [50,51,52]. In short, the 4 dpf (days post fertilization) period zebrafish were randomly divided into 6 groups (thirty fishes, *n* = 30), which were control group, *t*-BHP group, GSH (200 μM) group, and three TSE treatment (5, 10 and 20 μg/mL) groups. TSE groups were exposed to different concentration of TSE, while GHS groups received GSH (200 μM). All zebrafish were treated with *t*-BHP (2 mM) simultaneously, except the control groups. After treatment, the zebrafish were incubated for 48 h. Similarly, for the flavonoids in TSE, the zebrafish larvae were administrated with respective compounds (10 μM).

#### 4.2.9. Measurement of the Intracellular ROS Generation and Image Analysis in Zebrafish

The ROS level was measured by the reactive oxygen species assay kit. The zebrafish larvae were collected and incubated with 10 μM of DCFH-DA for 2 h. Afterwards, the zebrafish larvae were washed thrice by the embryo culture medium to remove the surficial fluorescent dyes. The fluorescence of zebrafish larvae was observed and photographed using a fluorescence inverted microscope (Carl Zeis, Germany). The fluorescence intensity of individual larva was quantified using fluorescent microplate reader (Bio Tek Synergy H1, Vermont, Winooski, USA).

#### 4.2.10. Statistical Analysis 

Data were analyzed by IBM SPSS Statistics 25.0 (IBM Corporation, Armonk, NY, USA), GraphPad Prism 8.0, and Excel 2010 and expressed as means ± SD. *p* values were determined by one-way ANOVA followed by LSD multiple comparisons test. *p* < 0.05 was considered to be statistically significant, *p* < 0.001 was considered to be extremely significant.

## 5. Conclusions

Tamarind shell extract (TSE) was rich in flavonoids, the major components were (+)-catechin, taxifolin, myricetin, eriodictyol, luteolin, morin, apigenin and naringenin. TSE manifested potent antioxidative capacity in vitro and in vivo. TSE could not only react with free radical directly to reduce the accumulation of lipid peroxidation products, but also reinstate the antioxidative defense system within cells by recovering the SOD activity. Importantly, the antioxidative benefits of TSE were associated with the flavonoids, especially eriodictyol, morin and taxifolin. To sum up, TSE has a strong ability to ameliorate the oxidative stress and related disorders and has the potential to be developed as functional products or food additives in future. 

## Figures and Tables

**Figure 1 molecules-28-01885-f001:**
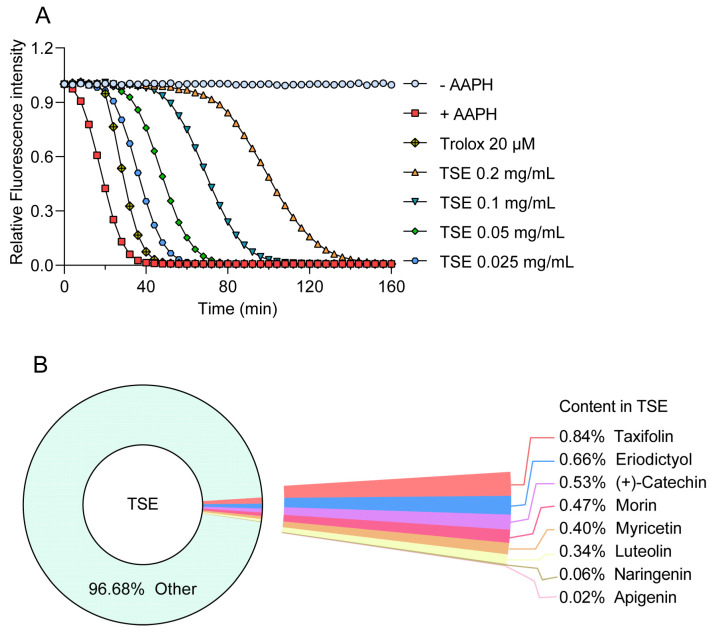
Antioxidant capacity of TSE in vitro evaluated by the ORAC assay. (**A**) The fluorescence attenuation curve of TSE. *n* = 6. (**B**) The contents of 8 flavonoids in TSE.

**Figure 2 molecules-28-01885-f002:**
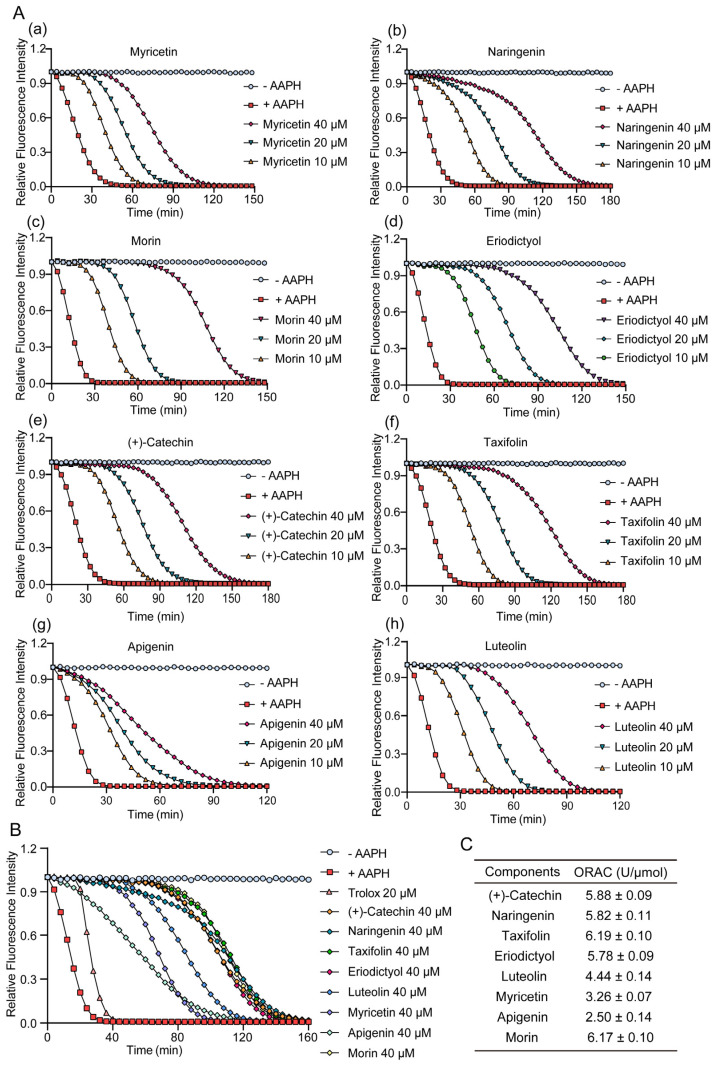
Inhibition effects of eight flavonoids on fluorescence decay in vitro. (**A**) Fluorescence signal attenuation curves of eight flavonoids in TSE, from (**a**–**h**) as followed myricetin, naringenin, morin, eriodictyol, (+)-catechin, taxifolin, apigenin, luteolin and respectively. *n* = 6. (**B**,**C**) The fluorescence signal attenuation curves and the ORAC values of 8 flavonoids at concentration of 40 μM. *n* = 6. The ROS scavenging capacity of eight compounds were detected by ORAC, using AAPH as a ROS generator, and the antioxidative activity was calculated as ORAC (U/μmol).

**Figure 3 molecules-28-01885-f003:**
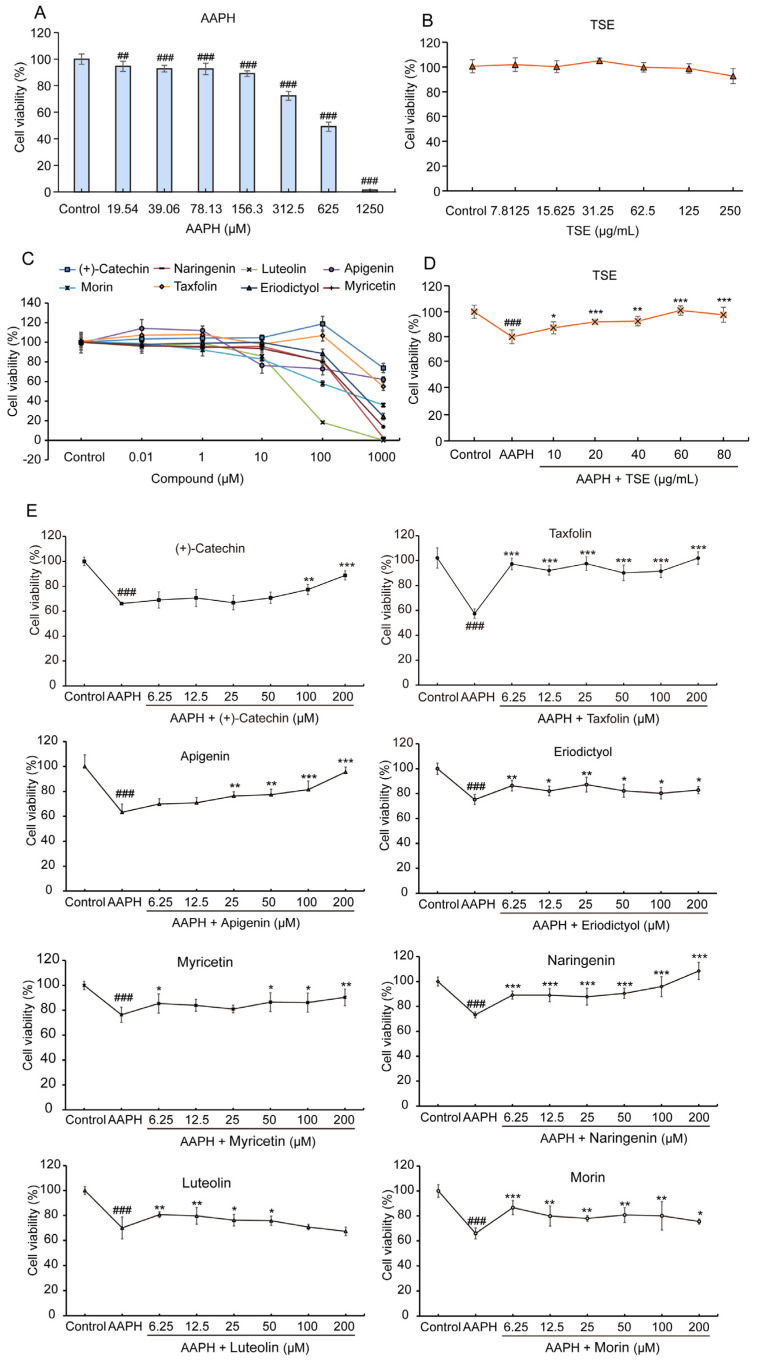
Effects of TSE and its eight flavonoids on AAPH-treated ATDC5 cell. (**A**–**C**) The cytotoxicity of AAPH, TSE and eight flavonoids in ATDC5 cells. *n* = 6. (**D**,**E**) Protective effect of TSE and its eight flavonoids against AAPH (400 μM)-induced cell death in ATDC5 cells. *n* = 4–6. Data were expressed as mean ± SD and the statistical differences were analyzed by one-way ANOVA followed by LSD multiple comparisons test (IBM SPSS Statistics 25.0, Armonk, NY, USA). *^##^ p* < 0.01, *^###^ p* < 0.001 vs. the control group. * *p* < 0.05, ** *p* < 0.01, *** *p* < 0.001 vs. AAPH group.

**Figure 4 molecules-28-01885-f004:**
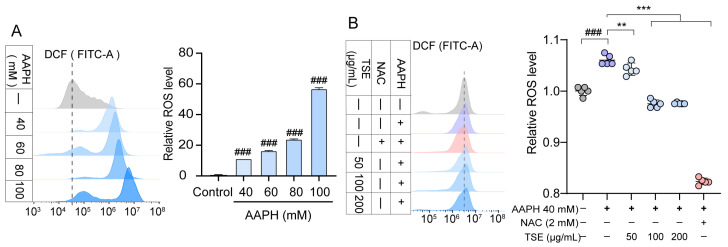
Effects of TSE on oxidative stress in AAPH-treated ATDC5 cells. (**A**) Oxidative stress in ATDC5 cells treated with AAPH (40–100 mM). *n* = 5; (**B**) ROS in AAPH-treated cells treated with the TSE (50–200 μg/mL). NAC (2 mM) acts as a positive control. *n* = 5. Data were expressed as mean ± SD and the statistical differences were analyzed by one-way ANOVA followed by LSD multiple comparisons test (IBM SPSS Statistics 25.0, Armonk, NY, USA). *^###^ p* < 0.001 vs. the control group. ** *p* < 0.01, *** *p* < 0.001 vs. AAPH group.

**Figure 5 molecules-28-01885-f005:**
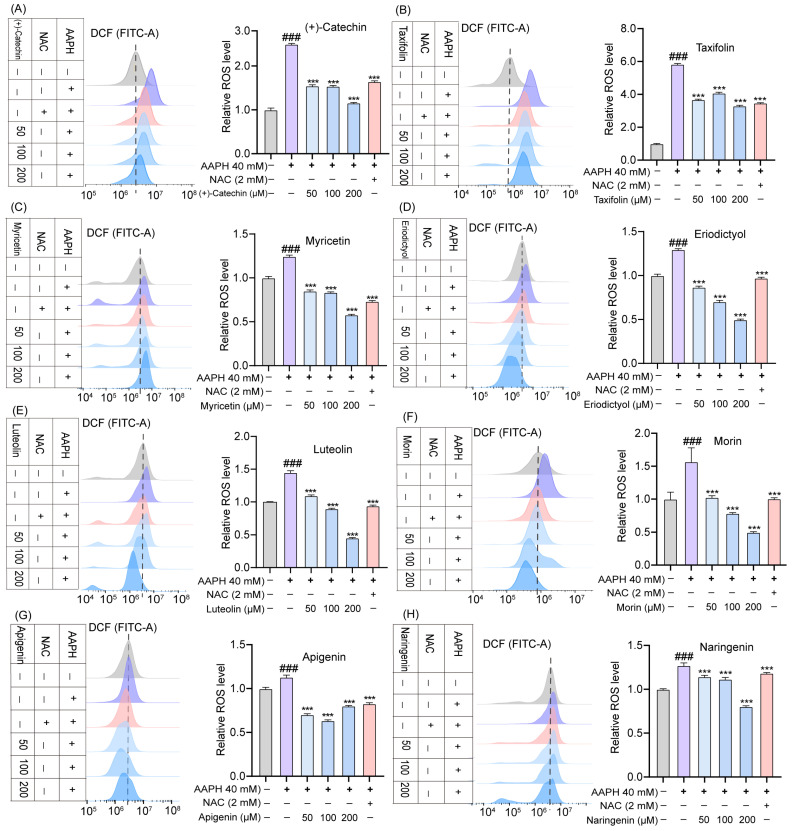
(**A**–**H**) ROS levels in AAPH-treated ATDC5 cells treated with the flavonoids of TSE (50–200 μM). NAC (2 mM) acts as a positive control. *n* = 5. Data were expressed as mean ± SD and the statistical differences were analyzed by one-way ANOVA followed by LSD multiple comparisons test (IBM SPSS Statistics 25.0, Armonk, NY, USA). *^###^ p* < 0.001 vs. the control group. *** *p* < 0.001 vs. AAPH group.

**Figure 6 molecules-28-01885-f006:**
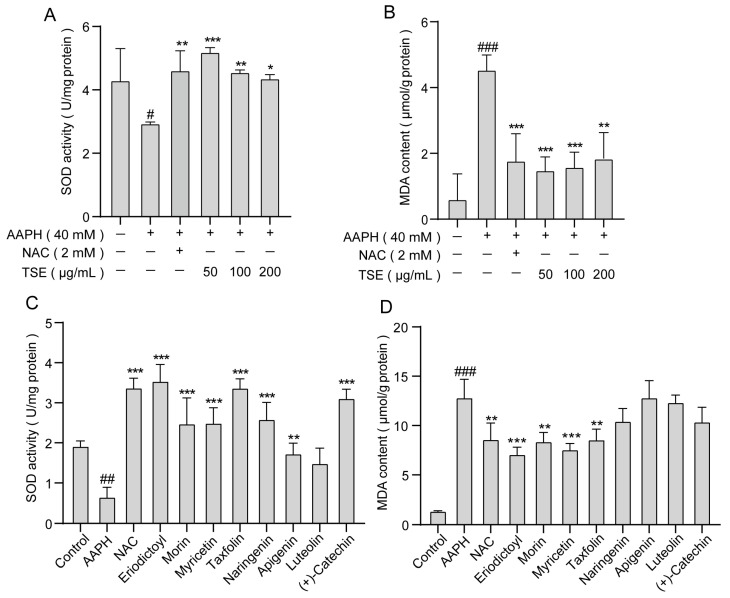
Effects of TSE and its flavonoids on the SOD activity and MDA content in the AAPH-treated ATDC5 cells. (**A**,**B**) The SOD activity and MDA content in ATDC5 cells treated with the TSE for 2 h, and co-incubated with 40 mM AAPH for 1 h. *n* = 3. (**C**,**D**) The SOD activity and MDA content in ATDC5 cells treated with eight flavonoids for 2 h, and co-incubated with 40 mM AAPH for 1 h. *n* = 3. NAC (2 mM) acts as a positive control. Data were expressed as mean ± SD and the statistical differences were analyzed by one-way ANOVA followed by LSD multiple comparisons test (IBM SPSS Statistics 25.0, Armonk, NY, USA). ^#^
*p* < 0.05, ^##^
*p* < 0.01, ^###^
*p* < 0.001 vs. the control group. * *p* < 0.05, ** *p* < 0.01, *** *p* < 0.001 vs. AAPH group.

**Figure 7 molecules-28-01885-f007:**
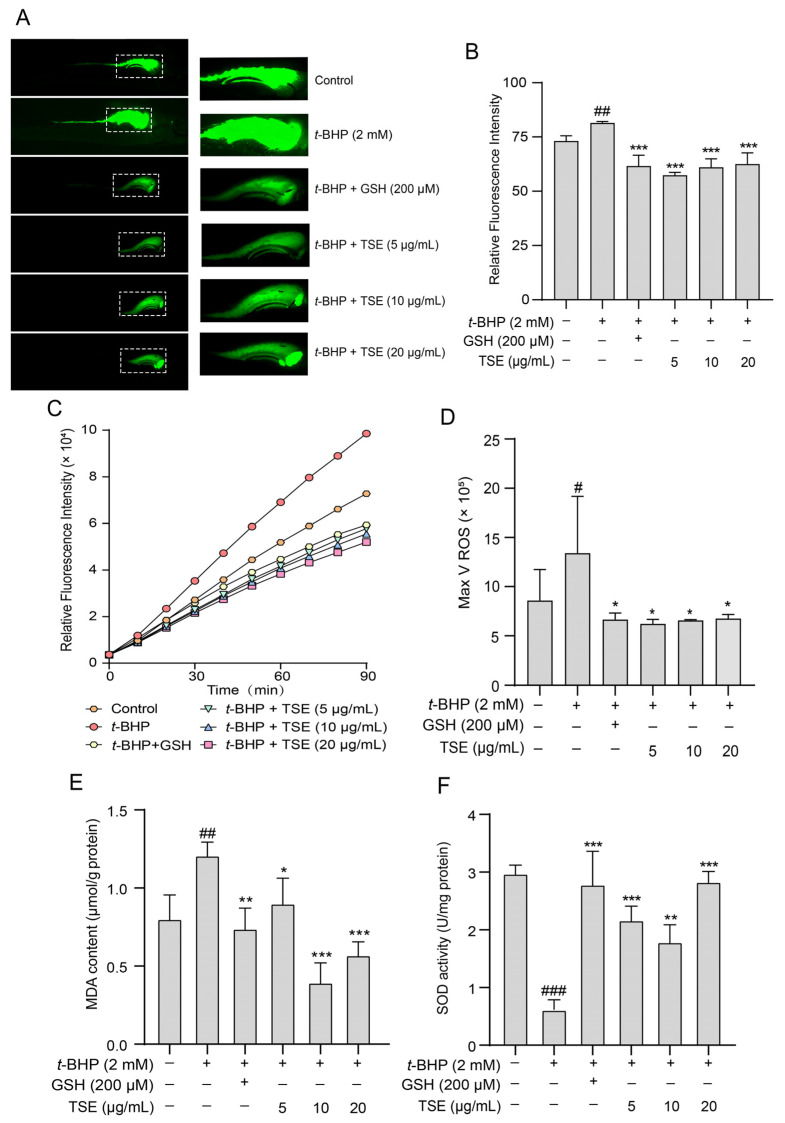
Effect of TSE on *t*-BHP-induced oxidative stress in zebrafish. (**A**) The fluorescence intensity of DCF in zebrafish co-treated with TSE (5–20 μg/mL) and *t*-BHP (2 mM) for 48 h (50× magnification). *n* = 5. (**B**) The quantitative analysis of fluorescence intensity in (**A**). (**C**) The ROS generation in zebrafish co-treated with TSE (5–20 μg/mL) and *t*-BHP (2 mM) for 48 h was monitored using a fluorescence microplate reader. *n* = 3. (**D**) The analysis of ROS maximum generation rate in (**C**). (**E**,**F**) The MDA content and SOD activity in zebrafish co-treated with TSE (5–20 μg/mL) and *t*-BHP (2 mM) for 48 h. *n* = 3. GSH (200 μM) acts as a positive control. Data were expressed as mean ± SD and the statistical differences were analyzed by one-way ANOVA followed by LSD multiple comparisons test (IBM SPSS Statistics 25.0, Armonk, NY, USA). ^#^
*p* < 0.05, ^##^
*p* < 0.01, ^###^
*p* < 0.001 vs. the control group. * *p* < 0.05, ** *p* < 0.01, *** *p* < 0.001 vs. *t*-BHP group.

**Figure 8 molecules-28-01885-f008:**
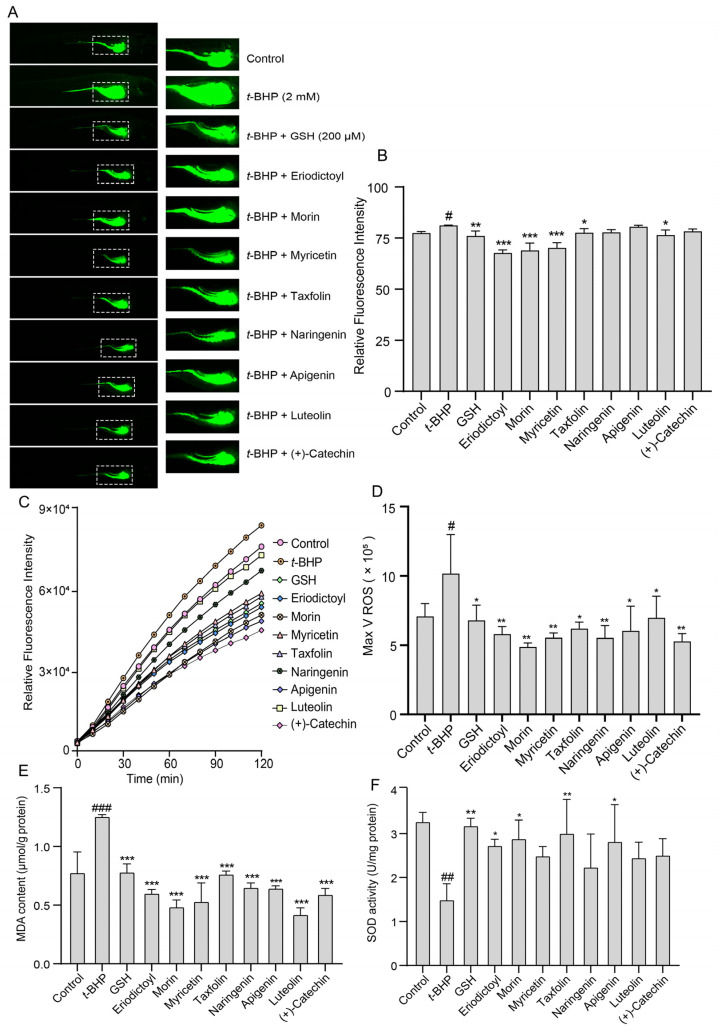
Effect of eight flavonoids on *t*-BHP-induced oxidative stress in zebrafish. (**A**) The fluorescence intensity in zebrafish co-treated with eight flavonoids (10 μM) and *t*-BHP (2 mM) for 48 h (50× magnification). (**B**) The quantitative analysis of fluorescence intensity in (**A**). (**C**) The ROS generation was monitored using a fluorescence microplate reader in zebrafish co-treated with eight flavonoids (10 μM) and *t*-BHP (2 mM) for 48 h. (**D**) The analysis of ROS maximum generation rate in (**C**). (**E**,**F**) The MDA content and SOD activity in zebrafish co-treated with eight flavonoids and *t*-BHP for 48 h (*n* = 3). GSH (200 μM) acts as a positive control. Data were presented as mean ± SD and the statistical differences were analyzed by one-way ANOVA followed by LSD multiple comparisons test (IBM SPSS Statistics 25.0, Armonk, NY, USA). ^#^
*p* < 0.05, ^##^
*p* < 0.01, ^###^
*p* < 0.001 vs. the control group. * *p* < 0.05, ** *p* < 0.01, *** *p* < 0.001 vs. *t*-BHP group.

**Table 1 molecules-28-01885-t001:** Flavonoids identified in TSE.

Compounds	Molecular Formula	Molecular Weight (g/mol)	Structures	Contents (mg/g)
(+)-Catechin	C_15_H_14_O_6_	290.08	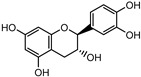	5.287
Taxifolin	C_15_H_12_O_7_	304.06	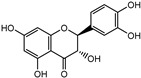	8.419
Eriodictyol	C_15_H_12_O_6_	288.06	** 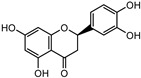 **	6.583
Morin	C_15_H_10_O_7_	302.04	** 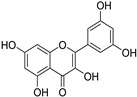 **	4.651
Myricetin	C_15_H_10_O_8_	318.04	** 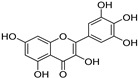 **	4.042
Luteolin	C_15_H_10_O_6_	286.05	** 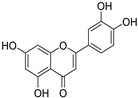 **	3.421
Naringenin	C_15_H_12_O_5_	272.07	** 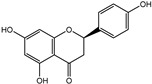 **	0.623
Apigenin	C_15_H_10_O_5_	270.05	** 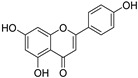 **	0.203

## Data Availability

The data presented in this study are available on request from the corresponding author.

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
