# Peer review of "Potential of Tamarind Shell Extract against Oxidative Stress In Vivo and In Vitro"

_molecules, 2023, doi:10.3390/molecules28041885_

Round 1

Reviewer 1 Report

Please provide details for section "4.1.1 Plant materials", such as how to obtain, identified and authenticated by whom? google maps coordinates must be included, etc.

Also provide details for the method "4.2.7 Measurement of SOD activity and MDA content in ATDC5 cells", "4.2.10 Measurement of the total SOD activity and MDA contents in zebrafish" are all too short not to even refer to any references.

Ethical approval for use in experimental animals must be given to this manuscript.

what does "to maintain the circadian rhythms" mean on lin 309? whereas there is no observation of circadian rhythms markers. Be clear.

Please authors provide an abbreviated section of list

Author Response

We are grateful for your positive comments and constructive feedback. The manuscript has been exactly revised according to your suggestions point by point. All the modifications have been highlighted in red in the revised manuscript.

Comments and response:

  1. Please provide details for section "4.1.1 Plant materials", such as how to obtain, identified and authenticated by whom? google maps coordinates must be included, etc.

Response: Thank you for your comment. The information about tamarind mentioned above has been added in the revised manuscript (Line 308-311).

  1. Also provide details for the method "4.2.7 Measurement of SOD activity and MDA content in ATDC5 cells", "4.2.10 Measurement of the total SOD activity and MDA contents in zebrafish" are all too short not to even refer to any references.

Response: Thank you for your comment. The experimental processes have been described in detail about measurement of SOD activity and MDA content in ATDC5 cells and zebrafish (Line 399-414).

  1. Ethical approval for use in experimental animals must be given to this manuscript.

Response: The ethical approval of zebrafish has been declared in the revised manuscript as follows: All the zebrafish used in current study are housed in an AAALAC-accredited facility. All experiments were carried out according to standardized protocols and approved by the Animal Care and Use Committee of Yunnan University of Chinese Medicine (Line 330-332).

  1. what does "to maintain the circadian rhythms" mean on line 309? whereas there is no observation of circadian rhythms markers. Be clear.

Response: Thank you for your comment. In fact, we mean light rhythm with a cycle of 14 h light and 10 h darkness. We modify these confusing words (Line 327).

  1. Please authors provide an abbreviated section of list.

Response: Thank you for your advice. We provide a list of abbreviations, as shown on Line 448-454.

Reviewer 2 Report

The research article titled” Potential of tamarind shell extract against oxidative stress in in vivo and in vitro” is a detailed study focusing on antioxidant properties of tamarind shell extract (TSE). The authors have shown that the TSE has various flavonoids isolated from TSE have capabilities to reduce oxidative stress induced by oxidative agents in vitro and in vivo. This is through paper with well thought our experimentation.  

The paper can be improved significantly by including the changes below:

1.     One of the major concerns for the paper is the significance of TSE in comparison to tamarind seed extract. The readers will benefit if the effect of TSE is comparable or superior to tamarind seed extract.

2.     Please clarify how the endogenous levels of each of the flavonoid is comparable to the dosage that has been used to show ROS scavenging in Fig 1.

3.     In Fig 3 E. Curve showing cell death induced AAPH must be included to see the protection offered by flavonoids from AAPH.

4.     In fig 5g, luteolin is offering protection at 100 uM but similar concentration of Luteolin in Fig 3 shows cell killing. Please clarify.

5.     It will help the reader to discuss in the discussion section why tamarind shell extract has these unique protective flavonoids and why a study on TSEThe research article titled” Potential of tamarind shell extract against oxidative stress in in vivo and in vitro” is a detailed study focusing on antioxidant properties of tamarind shell extract (TSE). The authors have shown that the TSE has various flavonoids isolated from TSE have capabilities to reduce oxidative stress induced by oxidative agents in vitro and in vivo. This is through paper with well thought our experimentation.  

The paper can be improved significantly by including the changes below:

1.     One of the major concerns for the paper is the significance of TSE in comparison to tamarind seed extract. The readers will benefit if the effect of TSE is comparable or superior to tamarind seed extract.

2.     Please clarify how the endogenous levels of each of the flavonoid is comparable to the dosage that has been used to show ROS scavenging in Fig 1.

3.     In Fig 3 E. Curve showing cell death induced AAPH must be included to see the protection offered by flavonoids from AAPH.

4.     In fig 5e, luteolin is offering protection at 100 uM but similar concentration of Luteolin in Fig 3c shows cell killing. Please clarify.

5.     It will help the reader to discuss in the discussion section why tamarind shell extract has these unique protective flavonoids and why a study on TSE is an important step to understand the potential of natural products as nutraceuticals.

Minor edits:

Line 411 can be rewritten as “TSE has the potential to be developed as …..

Author Response

Review 2

We are grateful for your positive comments and constructive feedback. The manuscript has been exactly revised according to your suggestions point by point. All the modifications have been highlighted in red in the revised manuscript.

Comments and response:

  1. One of the major concerns for the paper is the significance of TSE in comparison to tamarind seed extract. The readers will benefit if the effect of TSE is comparable or superior to tamarind seed extract.

Response: Thank you for your comments. I am sorry that we did not compare the activity of tamarind shells extract with tamarind seeds extract. Tamarind seeds are already widely studied and applied as some value-added products, while tamarind shells are usually abandoned as by-products. Our study confirmed antioxidant potential of tamarind shells due to the variety of active ingredients. We hope to enhance the utilization of tamarind shells as antioxidant supplements for higher additional value, and to reduce the cost of handling tamarind shells as waste (Line 71-75; 81-83; 244-248; 445-447).

  1. Please clarify how the endogenous levels of each of the flavonoid is comparable to the dosage that has been used to show ROS scavenging in Fig 1.

Response: Thank you for your suggestion. We detected the content of various flavonoids in tamarind shells extract (Table 1). According to your suggestion, we also clarify the levels of each of the flavonoid in figure 1B (Line 105-106).

  1. In Fig 3E. Curve showing cell death induced AAPH must be included to see the protection offered by flavonoids from AAPH.

Response: Thank you for your suggestion. In fact, a single dose of AAPH is applied in figure 3e, and AAPH-induced cell death is shown in the second column. We improved the picture to make it clearer (figure 3E) (Line 129-130).

  1. In fig 5g, luteolin is offering protection at 100 μM but similar concentration of Luteolin in Fig 3 shows cell killing. Please clarify.

Response: Thank you for your comment. In AAPH-induced oxidative stress experiment (figure 5g), ATDC5 cells were pretreated with luteolin for 2 h and then with AAPH for addition 1 h. The ROS levels were immediately detected by flow cytometry. However, in the cell viability experiment (figure 3c), ATDC5 cells were treated for 24 h. Hence, we think that luteolin has intracellular antioxidant activity to inhibit the increase in ROS induced by AAPH in the short term, but the stimulation for a longer time can lead to cytotoxicity through other underlying mechanisms. We have clarified this in the revised manuscript (Line 147-152).

  1. It will help the reader to discuss in the discussion section why tamarind shell extract has these unique protective flavonoids and why a study on TSE is an important step to understand the potential of natural products as nutraceuticals.

Response: Thank you very much for this excellent advice. We have highlighted the effects of protective flavonoids in tamarind shells against oxidizing agent and the importance of current research for development of tamarind shell extract into food or nutraceutical products in future. The supplementary content is detailed in the discussion section (Line71-75; Line81-83).

  1. Line 411 can be rewritten as “TSE has the potential to be developed as …..

Response: Thank you very much. We have revised this sentence according to your suggestion (Line445-447).

Reviewer 3 Report

Dear Editors, dear Authors,

I reviewed the article entitled "Potential of tamarind shell extract against oxidative stress in vivo and in vitro", written by Wei-xi Li, Rong-ping Huang, Shao-cong Han, Xi-you Li, Hai-biao Gong, Qiong-yi Zhang, Chang-yu Yan, Yi-fang Li and Rong-rong He.

The purpose of this manuscript is to analyse the chemical composition of tamarind pell extract and to assess its antioxidant capability in vitro and in vivo.

The scientific collect is very interesting. However, some problems, as indicated below, should be addressed before the document can be considered for publication in this journal. This version of the manuscript is not enough complete. Here, I present all my objections in detail.

Minor revision

English language and style are not exhaustive, a greater spell check is required to ensure that an international audience can clearly understand your text. In general, I suggest reviewing the style of the manuscript according to the guidelines of the journal.

Abstract: The abstract is exhaustive in all its parts.

Introduction:

·       Line 31: I suggest to specify better the concept of redox imbalance.

·       Line 45: The fact that the tamarind is rich in fructose may contrast with the paper’s objective, because of its oxidizing power witnessed by scientific literature. I suggest to include another example of the nutrient contained in the tamarind or to indicate the concentration of fructose present in the fruit.

·       Line 50: The presence of different phytochemicals, such as polyphenols, in the seed but also in the tamarind fruit justifies its use as an antioxidant compound. Indeed, the strong antioxidant activity of polyphenols against various stresses is widely demonstrated in scientific literature. I suggest to emphasize this concept by adding the following references: doi: 10.3390/antiox10020283; doi: 10.3390/ijms231910991; doi:10.1155/2021/9932218.

·       Lines 56-62: The fact that tamarind peel could pollute the environment and that its alternative use as an antioxidant could prevent this waste is extremely interesting. I suggest to emphasize this concept.

·       Line 64: Add the references of the sentence " Previous studies have reported that TSE could scavenge free radicals in vitro".

Results:

·       Line 84, Table 1: Specify the unit of measurement of the molecular weight.

·       Line 86: Insert a short sentence on the principle of the method.

·       Line 94: Specify the number of experiments, the use of +AAPH as positive control and statistically significant differences between +AAPH and the tested substances.

·       Line 102: Specify the number of experiments.

·       Line 116, Figures 3A and 3C: Specify the statistically significant differences between AAPH and the control.

·       Line 122: Specify the number of experiments.

·       Lines 131-135: These sentences should be moved to line 126.

·       Line 136: “ROS” should be modified in “ATDC5 cells”.

·       Line 141: “AAPH-induced cells” should be modified in “AAPH-treated ATDC5 cells”.

·       Line 147: Specify what is the model group.

·       Line 157:AAPH induced ATDC5 cells” should be modified in “AAPH-treated ATDC5 cells”.

·       Line 159: What is 40 mM?

Discussion:

·       Line 216: I suggest to write a short sentence that contains some examples of pathologies related to increased oxidative stress.

Material and methods:

·       Line 312: It is preferable to express the salt concentration in mM.

·       Line 316: What is the degree of fruit ripeness? At what time of year was it harvested?

·       Line 341: I suggest to briefly describe the method.

·       Line 351: Specify the company that supplied the FBS.

·       Line 378: I suggest to briefly describe the method.

·       Line 399: I suggest to briefly describe the method.

Conclusion:

·       What is the practical significance of this scientific work? Could the extract be useful in counteracting the pathologies induced by oxidative stress?

References:

·       In general, there are few bibliographical references to support this work. Therefore, I suggest implementing them.

Major revision:

·       Looking at the statistically significant differences between the various graphs, something does not convince me, especially in Figure 5. To make the data more reliable, I recommend the authors apply a post hoc test (Dunnett or Bonferroni, depending on the case) after the ANOVA and specify it in the caption of each figure.

Author Response

Review 3

We are grateful for your positive comments and constructive feedback. The manuscript has been exactly revised according to your suggestions point by point. All the modifications have been highlighted in red in the revised manuscript.

Comments and response:

  1. English language and style are not exhaustive, a greater spell check is required to ensure that an international audience can clearly understand your text. In general, I suggest reviewing the style of the manuscript according to the guidelines of the journal.

Response: Thank you for your suggestion. We checked the whole manuscript carefully and improved English language for reader understanding. In addition, we revised the manuscript according to the guidelines of the journal. 

  1. The abstract is exhaustive in all its parts.

Response: Thank you for your comment.

  1. Line 31: I suggest to specify better the concept of redox imbalance.

Response: Thank you for your excellent advice. We modified the introduction of redox balance/imbalance concept, Line 32-42 for details.

  1. Line 45: The fact that the tamarind is rich in fructose may contrast with the paper’s objective, because of its oxidizing power witnessed by scientific literature. I suggest to include another example of the nutrient contained in the tamarind or to indicate the concentration of fructose present in the fruit.

Response: Thank you for your suggestion. In fact, the fruit of tamarind contains fructose mentioned in the original manuscript, which does not contrast with the objective of our current study that shell of tamarind acts as an excellent antioxidant (Line 56).

  1. Line 50: The presence of different phytochemicals, such as polyphenols, in the seed but also in the tamarind fruit justifies its use as an antioxidant compound. Indeed, the strong antioxidant activity of polyphenols against various stresses is widely demonstrated in scientific literature. I suggest to emphasize this concept by adding the following references: doi: 10.3390/antiox10020283; doi: 10.3390/ijms231910991; doi:10.1155/2021/9932218.

Response: Thank you for your advice. We have highlighted the role of polyphenols in antioxidant effect of tamarind and cited these three references (Line 63).

  1. Lines 56-62: The fact that tamarind peel could pollute the environment and that its alternative use as an antioxidant could prevent this waste is extremely interesting. I suggest to emphasize this concept.

Response: Thank you for your advice. We have emphasized this concept by further illustrating this fact and citing some relevant references (Line 71-75).

  1. Line 64: Add the references of the sentence "Previous studies have reported that TSE could scavenge free radicals in vitro".

Response: Related references have been added (Line 77).

  1. Line 84, Table 1: Specify the unit of measurement of the molecular weight.

Response: The unit has been specified (Table 1).

  1. Line 86: Insert a short sentence on the principle of the method.

Response: The principle of ORAC assay has been added (Line 360-363).

  1. Line 94: Specify the number of experiments, the use of +AAPH as positive controland statistically significant differences between +AAPH and the tested substances.

Response: Thank you for your suggestion. The number of experiments has been specified. In fact, AAPH was used as radical initiator, and +AAPH group have no ORAC value without any antioxidant. Therefore, +AAPH group did not act as positive control. It isn’t necessary to analyze the significant differences between +AAPH group and tested substances groups.

  1. Line 102: Specify the number of experiments.

Response: The number of experiments has been specified (Line 112).

  1. Line 116,Figures 3A and 3C: Specify the statistically significant differences between AAPH and the control.

Response: The significant differences have been specified (Line 129).

  1. Line 122: Specify the number of experiments.

Response: The number of experiments has been specified (Line 132).

  1. Lines 131-135: These sentences should be moved to line 126.

Response: Thanks for your suggestion. We have made adjustment (Line 139-143).

  1. Line 136: “ROS” should be modified in “ATDC5 cells”.

Response: Thanks for your comment. We have revised it (Line 155).

  1. Line 141: “AAPH-induced cells” should be modified in “AAPH-treated ATDC5 cells”.

Response: Thanks for your comment. We have revised it (Line 161).

  1. Line 147: Specify what is the model group.

Response: We have changed ‘model group’ into ‘AAPH group’ (Line 168).

  1. Line 157: “AAPH induced ATDC5 cells” should be modified in “AAPH-treated ATDC5 cells”.

Response: Thanks for your comment. We have revised it (Line 179).

  1. Line 159: What is 40 mM?

Response: The missing information has been added (Line 181).

  1. Line 216: I suggest to write a short sentence that contains some examples of pathologies related to increased oxidative stress.

Response: Thanks for your advice. We have added some widely reported pathologic conditions related to oxidative stress or damage (Line 250-252).

  1. Line 312: It is preferable to express the salt concentration in mM.

Response: We have expressed the concentration of salt in molarity units (Line 329-330).

  1. Line 316: What is the degree of fruit ripeness? At what time of year was it harvested?

Response: Thanks for your comments. Relevant information has been added in the methods section (Line 309-311).

  1. Line 341: I suggest to briefly describe the method.

Response: The relevant methods have been described (Line 360-363).

  1. Line 351: Specify the company that supplied the FBS.

Response: The supplier information of serum has been replenished (Line 373).

  1. Line 378: I suggest to briefly describe the method.

Response: The relevant methods have been described (Line 399-414).

  1. Line 399: I suggest to briefly describe the method.

Response: The relevant methods have been described (Line 399-414).

  1. What is the practical significance of this scientific work? Could the extract be useful in counteracting the pathologies induced by oxidative stress?

Response: Thank you for your advice. In the conclusion, we emphasized the potential application of tamarind shell extract in pathology related to oxidative stress in future (Line 442-447).

  1. In general, there are few bibliographical references to support this work. Therefore, I suggest implementing them.

Response: we have cited some bibliographical references to support our current work (Line57; Line63; Line 291-293).

  1. Looking at the statistically significant differences between the various graphs, something does not convince me, especially in Figure 5. To make the data more reliable, I recommend the authors apply a post hoc test (Dunnett or Bonferroni, depending on the case) after the ANOVA and specify it in the caption of each figure.

Response: Thank you for your excellent suggestion. We have conducted post hoc test after the one-way ANOVA on all the data which need multiple group comparisons. The actual method used was indicated in the figure legends(Line 135; Line 158; Line 163-164; Line 184-185; Line 222-223; Line 233-234).

Round 2

Reviewer 3 Report

The authors made the suggested changes.